# Interplay between a polerovirus and a closterovirus decreases aphid transmission of the polerovirus

Souheyla Khechmar,[1,2] Quentin Chesnais,[1,2] Claire Villeroy,[1] Véronique Brault,[1,2] Martin Drucker[1,2]

**ABSTRACT** Multi-infection of plants by viruses is very common and can change drastically infection parameters such as virus accumulation, distribution, and vector transmission. Sugar beet is an important crop that is frequently co-infected by the polerovirus beet chlorosis virus (BChV) and the closterovirus beet yellows virus (BYV), both vectored by the green peach aphid (*Myzus persicae*). These phloem-limited viruses are acquired while aphids ingest phloem sap from infected plants. Here we found that co-infection decreased transmission of BChV by ~50% but had no impact on BYV transmission. The drastic reduction of BChV transmission was due to neither lower accumulation of BChV in co-infected plants nor reduced phloem sap ingestion by aphids from these plants. Using the signal amplification by exchange reaction fluorescent *in situ* hybridization technique on plants, we observed that 40% of the infected phloem cells were co-infected and that co-infection caused redistribution of BYV in these cells. The BYV accumulation pattern changed from distinct intracellular spherical inclusions in mono-infected cells to a diffuse form in co-infected cells. There, BYV co-localized with BChV throughout the cytoplasm, indicative of virus-virus interactions. We propose that BYV-BChV interactions could restrict BChV access to the sieve tubes and reduce its accessibility for aphids and present a model of how co-infection could alter BChV intracellular movement and/or phloem loading and reduce BChV transmission.

**IMPORTANCE** Mixed viral infections in plants are understudied yet can have significant influences on disease dynamics and virus transmission. We investigated how co-infection with two unrelated viruses, BChV and BYV, affects aphid transmission of the viruses in sugar beet plants. We show that co-infection reduced BChV transmission by about 50% without affecting BYV transmission, despite similar virus accumulation rates in co-infected and mono-infected plants. Follow-up experiments examined the localization and intracellular distribution of the viruses, leading to the discovery that co-infection caused a redistribution of BYV in the phloem vessels and altered its repartition pattern within plant cells, suggesting virus-virus interactions. In conclusion, the interplay between BChV and BYV affects the transmission of BChV but not BYV, possibly through direct or indirect virus-virus interactions at the cellular level. Understanding these interactions could be crucial for managing virus propagation in crops and preventing yield losses.

**KEYWORDS** plant virus, virus co-infection, virus localization, aphid transmission, virus accumulation, aphid behavior, plant-virus-virus-vector interactions

Historically, studies on diseases in both plants and animals focused mainly on simple interactions implicating only one host and one pathogen. However, in nature, a host is often subjected to a multitude of pathogens simultaneously, a scenario referred to as multi-infection or mixed infection, which might be the rule rather than the exception (1, 2). The development of molecular tools such as high-throughput sequencing further highlighted the high occurrence of mixed infections compared

Address correspondence to Véronique Brault, veronique.brault@inrae.fr, or Martin Drucker, martin.drucker@inrae.fr.

The authors declare no conflict of interest.

See the funding table on p. 16.

to mono-infection (2). Mixed infections can be heterogeneous (i.e., implying different pathogens such as bacteria, fungi, or viruses), or homogeneous (i.e., with similar microorganisms) (3). Multi-infection can occur through the simultaneous inoculation of several pathogens or sequentially, causing co-infection or super-infection, respectively. The outcome of a co-infection or super-infection will be affected by the intrahost interactions between the different infectious agents as well as by their interaction with the host or, in the case of vector-borne pathogens, with the vector (2).

Pathogen interactions, plant viruses in our case, within hosts can be synergistic; i.e., the presence of one pathogen facilitates the infection of one or more additional pathogens. For example, synergistic interactions between tomato chlorosis virus (ToCV) and tomato infectious chlorosis virus (TICV) led to an increased TICV accumulation in *Nicotiana benthamiana* (4). Interactions between pathogens are considered neutral when they do not, or only marginally, interfere with each other, as demonstrated, e.g., for tomatoes infected with three begomoviruses, tomato yellow mottle virus, tomato leaf curl Sinaloa virus, and tomato yellow leaf curl virus. There, the accumulation of each virus was not strongly affected by the presence of the others, although the triple infection induced stronger symptoms (5). Finally, in antagonistic pathogen interactions, the presence of one pathogen impacts the infection of the other pathogen negatively (1, 2, 6, 7). An example is the co-infection of two tobamoviruses, hibiscus latent Singapore virus (HLSV) and tobacco mosaic virus (TMV), in *N. benthamiana* plants where HLSV levels decreased, compared to mono-infection, whereas TMV levels remained almost unaltered (8). As in this former example and unlike many synergistic interactions, antagonisms tend to occur mainly between closely related viruses, inducing fitness costs for one or both competitors (7).

Virus-virus and virus-host interactions in a multi-infection context can impact different aspects of the virus cycle, such as virus replication, virus movement, and virus localization in plant cells or tissues, or by interfering with plant defenses (9, 10). The outcomes of the viral multi-infection not only can alter the fitness of co-existing viruses but also can have a direct impact on the fitness of the plant itself by affecting the severity of foliar symptoms, plant development, and plant chemical and physiological profiles (11). Plant traits modified by infection or multi-infection can have consequences on virus transmission because they can change the nutritional quality of the plant or modify visual traits and production of volatiles that play important roles in vector-plant interactions. These alterations of the multi-infected plants can therefore influence vector behavior and subsequently virus transmission (11–14). Studies on mixed viral infections with regard to vector interaction and transmission are scarce. Ontiveros et al. reported that tomatoes co-infected with the begomovirus tomato yellow leaf curl virus (TYLCV) and the crinivirus ToCV were more attractive for the whitefly vectors than ToCV mono-infected plants (15). Co-infection with the reovirus raspberry latent virus and the closterovirus raspberry leaf mottle virus changed aphid vector preferences for infected raspberry plants but not their feeding behavior (16).

The viral and host factors mediating these synergistic or antagonistic effects have only been identified in a few cases. For instance, the synergistic interaction between the crinivirus sweet potato feathery mottle virus (SPCSV) and the potyvirus sweet potato chlorotic stunt virus is mediated by the SPCSV-encoded RNase3 protein (17). RNase3 functions as a viral suppressor of RNA silencing that might target a specific host component by interfering with small-RNA biogenesis (18). Another example is the potyviral silencing suppressor HC-Pro that increases the titer of potato leafroll virus (PLRV) in co-infection (19). One example on the host side is the salicylic acid-responsive gene PR-P6, whose expression correlates with virus titer and symptoms severity in tomatoes mono- or co-infected with TYLCV and ToCV (15).

Sugar beet (*Beta vulgaris*) is an economically important crop that is often co-infected in the field by several viruses (20, 21), causing alone, or in combination, leaf yellowing, yield reduction, and low sugar content. Among the viruses detected are the poleroviruses beet Western yellows virus (BWYV-USA) so far not reported in Europe, beet mild

yellowing virus and beet chlorosis virus (BChV), the closterovirus beet yellows virus (BYV), and a potyvirus, beet mosaic virus (BtMV), all occurring worldwide (20, 22). The impact of multiple sugar beet infections on the severity of leaf yellowing has been studied for BYV, BtMV, and BWYV-USA (21). In this study, sugar beet lines were inoculated with either one, two, or all three viruses. Faster appearance of symptoms and more severe stunting were observed in mixed infections with BYV and BtMV, compared to mono-infections (21). Acceleration of symptom development and greater yield losses were observed in sugar beets co-inoculated with BChV and BYV (20).

In this work, we focused on the co-infection of sugar beet with BYV and BChV. BYV is mainly transmitted by the two aphid species *Myzus persicae* and *Aphis fabae* (23). The main vector species for BChV is *M. persicae*, but this virus can also be transmitted efficiently by *Macrosiphum euphorbiae* (24, 25). BChV and BYV use two different modes of transmission: circulative and persistent for BChV, and non-circulative and semi-persistent for BYV. Viruses transmitted in the persistent and circulative modes are acquired during vector feeding and pass through the insect's digestive tract to be released into the hemolymph. From there, they reach the salivary glands, where they accumulate and are inoculated as a saliva component when aphids feed on new plants (26). This transmission mode is characterized by long phases of virus acquisition (hours to days) on the infected plant and latency (the time it takes for the virus to reach the aphid's salivary glands). The time required for aphids to inoculate the virus is also long (several hours), and aphids remain viruliferous for the rest of their lives (27). In the case of semi-persistent transmission, the virus attaches to areas of the mouthparts or anterior digestive tract. Maximum transmission is achieved after a few hours of acquisition phase, and aphids transmit viruses for only a few hours to a few days (27–29). The two viruses differ also in their cellular tropism. BChV is restricted to phloem tissues (30–32). BYV is also a phloem-limited virus but can colonize epidermal and mesophyll cells at a late stage of infection (33–35).

Here, we studied mixed viral infection of sugar beet and examined its effect on vector transmission. Our results show that co-infection of sugar beet with BChV and BYV had no impact on BYV transmission but did decrease transmission of BChV by 50%. We present evidence that the decrease in transmission is due to specific BYV-BChV interactions in co-infected cells and not to changes in virus accumulation or vector-feeding behavior on co-infected plants.

## RESULTS

### Virus transmission by aphids from sugar beet co-infected with BChV and BYV

We wanted to know whether virus acquisition from BYV/BChV co-infected source plants would affect the aphid transmission of each virus. For this, *Myzus persicae* aphids were allowed to acquire BChV and/or BYV from mono- or co-infected sugar beets and then transferred onto test plants for virus inoculation (Fig. 1). The BYV transmission rate was not affected by sugar beet co-infection (aphid transmission rate of 75% from mono-infected plants and 82% from BYV/BChV co-infected plants) (Fig. 1A). BChV transmission was significantly reduced when aphids fed on BYV/BChV co-infected plants compared to BChV mono-infected plants (30% vs 48%, respectively) (Fig. 1B).

Aphids feeding on co-infected plants transmitted both viruses together at 19% transmission rate, which was significantly lower than expected, based on the virus transmission efficiency of each virus [36% expected co-infection rate, calculated as the product of the transmission rates using mono-infected source plants; *P* value = 0.029; *n* = 75, five independent experiments; df = 1; chi-square ($\chi^2$), Fig. S1]. BYV alone was present in 63% of the aphid-inoculated plants, whereas BChV was detected alone in 11% of the infected plants (Fig. S1).

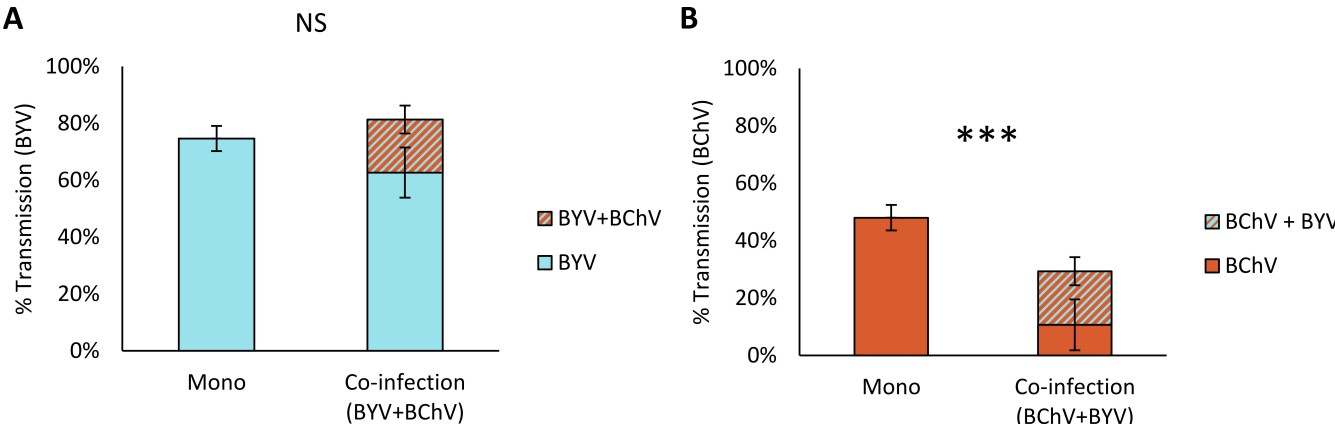

**FIG 1** Aphid transmission efficiency of (A) BYV and (B) BChV from mono- or co-infected plants. *Myzus persicae* acquired virus from mono or co-infected leaves for 24 h. Then, three aphids were transferred per test plant for 72-h inoculation. A double-antibody sandwich enzyme-linked immunosorbent assay was performed 3 weeks later to detect infection. The percentage of infected plants is indicated (% transmission). The difference was statistically significant for BChV but not for BYV transmission (generalized linear mixed effects model(GLMER), BYV: *P* value = 0.060; BChV: *P* value < 0.001; *n* = 75, five independent experiments; df = 1). NS, not significant, \*\*\**P* < 0.001.

## Aphid feeding behavior on mono-infected and co-infected plants or plants co-infected with both viruses

Aphid feeding behavior plays an important role in virus transmission. To address whether aphid feeding behavior was impacted on sugar beet co-infected with BChV and BYV, compared to mono-infected sugar beet, we used the electrical penetration graph (EPG) technique. Aphid total durations of probing time, intercellular pathway, intercellular salivation and the time spent to reach the phloem were the same for all conditions (Fig. 2A). However, aphids spent significantly more time (extra ~1 h) ingesting phloem sap (E2 phase) on BYV mono-infected than on healthy or BChV-infected plants. The total number of all feeding phases was not affected by the different conditions except for the number of extracellular salivations (E1e) that was significantly lower on BYV-infected plants than on BChV-infected plants; their number on healthy and BYV/BChV co-infected plants was intermediate (Fig. 2B).

## BChV and BYV accumulation in mono- or co-infected plants

To address whether BChV and BYV accumulation could be impacted by co-infection, we quantified virus titers by reverse transcription quantitative PCR (RT-qPCR) on sugar beet mono- or co-infected with BYV and BChV. To account for potential uneven virus distribution in the leaf, the entire leaf was ground and processed for RNA extraction and virus quantification. A reference gene, tubulin, was selected for calibration, among six tested, prior to virus quantification (Table S2). No significant differences in virus accumulation between mono-infected plants or BYV/BChV co-infected plants were observed (Fig. 3).

## Tissue distribution of BChV and BYV in mono- and co-infected plants

To assess if co-infection affected tissue localization of BChV or BYV, signal amplification by exchange reaction fluorescent *in situ* hybridization (SABER-FISH) was performed on mono- and co-infected plants at 21 days post-inoculation (dpi). In cross-sections of BYV mono-infected leaves, fluorescence corresponding to BYV genomes was observed exclusively in phloem cells as expected (Fig. 4B). The label was found in the phloem of primary (midribs) and secondary veins, and the label intensity, estimated by visual observations, was equal in both locations (Fig. 4B). In BChV mono-infected plants, fluorescence corresponding to BChV genomes was also observed, as expected, in the phloem of primary and secondary veins (Fig. 4C). However, the BChV label was less

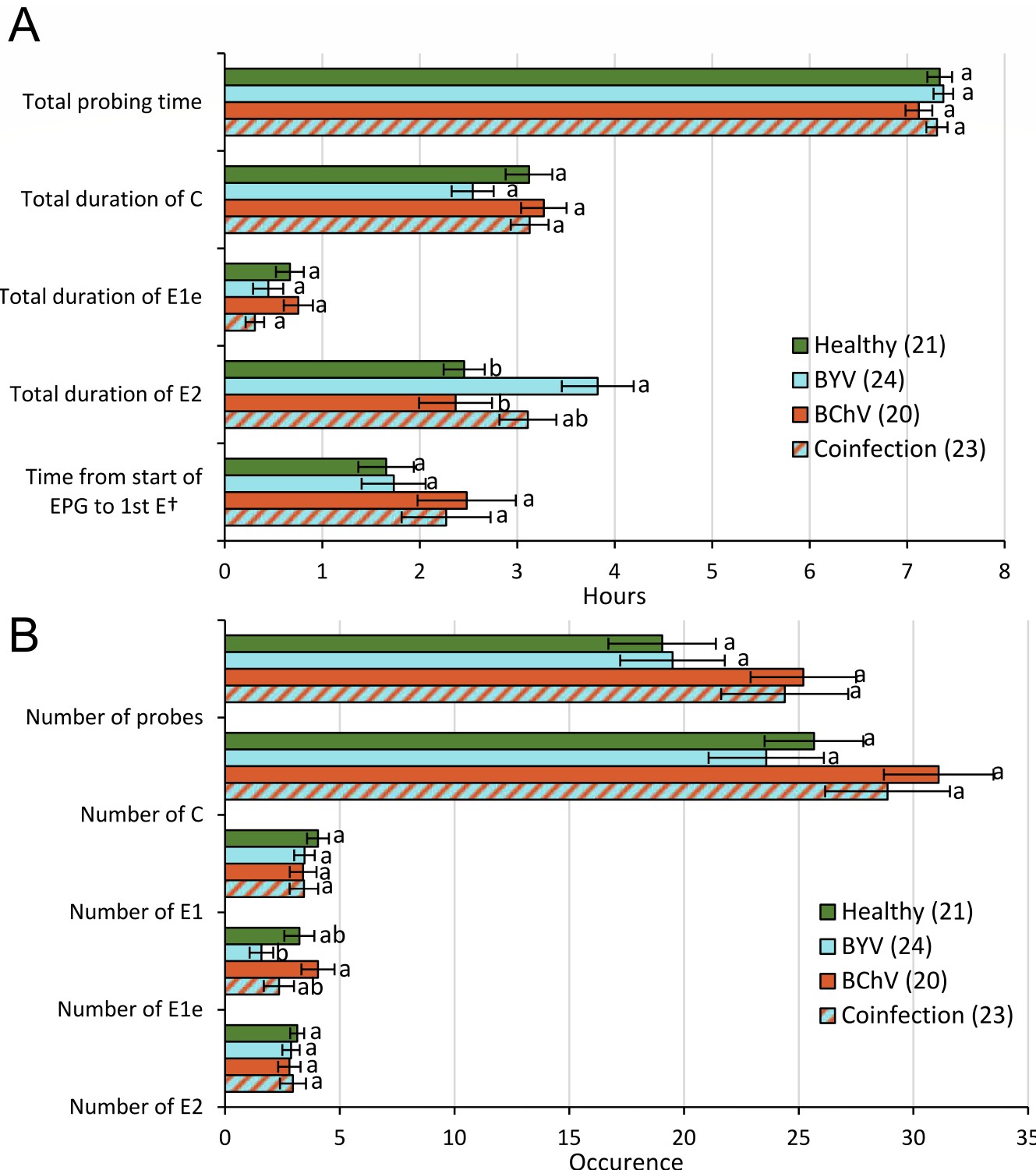

**FIG 2** Feeding behavior of *Myzus persicae* on healthy, BYV and BChV mono-infected, or BYV/BChV co-infected sugar beets. (A and B) The behavior of individual aphids was recorded by electrical penetration graph (EPG) for 8 h on the third upper leaf. Selected EPG parameters are presented according to (A) duration or (B) occurrence. The histogram bars display means and standard errors of the mean. Different letters indicate significant differences between plant infection status as tested by the generalized linear model (GLM) followed by pairwise comparisons using package R: emmeans ($P < 0.05$ method: Tukey honestly significant difference). Statistical analysis of the duration of the events indicates a significant difference for the duration of phloem sap ingestion (E2) on BYV-infected vs healthy or BChV-infected plants (GLM, df = 3, $\chi^2$ = 15.98, $P$ = 0.001) but no differences for the total duration of stylet penetrations (probing time) (GLM, df = 3, $\chi^2$ = 2.585, $P$ = 0.460), the total duration of pathway phases (C) (GLM, df = 3, $\chi^2$ = 6.962, $P$ = 0.073), the total duration of intercellular salivation (E1e) (GLM, df = 3, (Continued on next page)

Fig 2 (Continued)

$\chi^2$ = 5.645, $P$ = 0.130), and the time until the first sap ingestion from the phloem (Cox, df = 3, $P$ = 0.526). Statistical analysis of the occurrence of events revealed significant differences for the intercellular salivation between BYV- and BChV-infected plants (GLM, df = 3, $\chi^2$ = 8.581, $P$ = 0.035) but no differences for the number of stylet insertions (GLM, df = 3, $\chi^2$ = 4.725, $P$ = 0.193), pathway phases (C) (GLM, df = 3, $\chi^2$ = 4.954, $P$ = 0.175), salivation phases (E1) (GLM, df = 3, $\chi^2$ = 1.006, $P$ = 0.800), and phloem sap ingestions (GLM, df = 3, $\chi^2$ = 0.459, $P$ = 0.928). $n$ = 20–24 as indicated in the graphs.

intense in the primary veins than in the secondary veins. Co-infection altered the distribution of BYV and BChV (Fig. 4D). Compared to mono-infection, fewer cells were labeled with BYV in midribs and more cells in the secondary veins in co-infected plants. Compared to mono-infected plants, BChV was present in more phloem cells of primary veins in co-infected plants, while the BChV label in secondary veins was unchanged. Both viruses were still confined to phloem cells, and no escape in non-phloem cells was noticed (Fig. 4D through F).

We estimated the percentage of co-infected cells and mono-infected cells in co-infected plants. Ten to eleven sections from co-infected plants were observed in two independent experiments (Table 1). We noticed a slightly lower percentage of cells infected with BChV (63.2% of cells containing only BChV or co-infected) compared to BYV (74.3% of cells containing only BYV or co-infected) (Table 1). Between 30% and 50% of all infected cells were co-infected, and similar percentages of cells were mono-infected with BChV or BYV. This suggests that there is no exclusion between the two viruses at the cellular level. The label intensity of each virus seemed to be similar in mono-infected and co-infected cells, indicating that they did not interfere with each other's accumulation.

## Intracellular distribution of BYV and BChV in mono and co-infected plants

Cross sections of phloem tissue allow only a limited view of the phloem cell and sieve tube lumen because their elongated shapes are arranged perpendicular to the section plane. To better assess if co-infection affected intracellular distribution of BChV or BYV, SABER-FISH was performed on longitudinal sections of infected plants at 21 dpi (Fig. 5). In mono-infected plants, BYV label was present in infected cells as spherical cytoplasmic inclusions that aligned like pearl chains, similarly as reported before using light microscopy (36) (Fig. 5A). In the case of BChV mono-infected plants, the virus label was

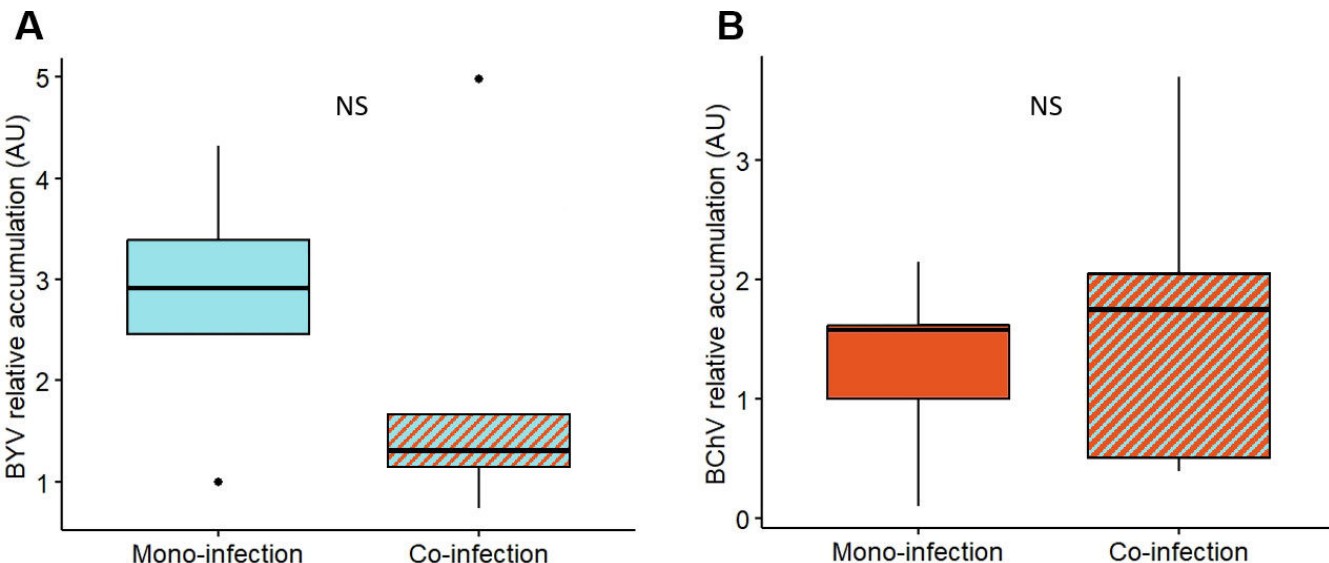

**FIG 3** Relative accumulation of BYV (A) and BChV (B) in mono- and co-infected plants at 21 dpi. Virus accumulation in source plants was measured by multiplex TaqMan RT-qPCR as described in Materials and Methods. Box plots show the median values (black lines), 25%–75% percentiles (boxes), 10%–90% percentiles (whiskers), and outliers (dots). Differences were not statistically significant (Student $t$-test, BYV: $t$ = −0.908, $P$ value = 0.390; BChV: $t$ = 0.59456, $P$ value = 0.569; $n$ = 5; df = 8). NS, not significant.

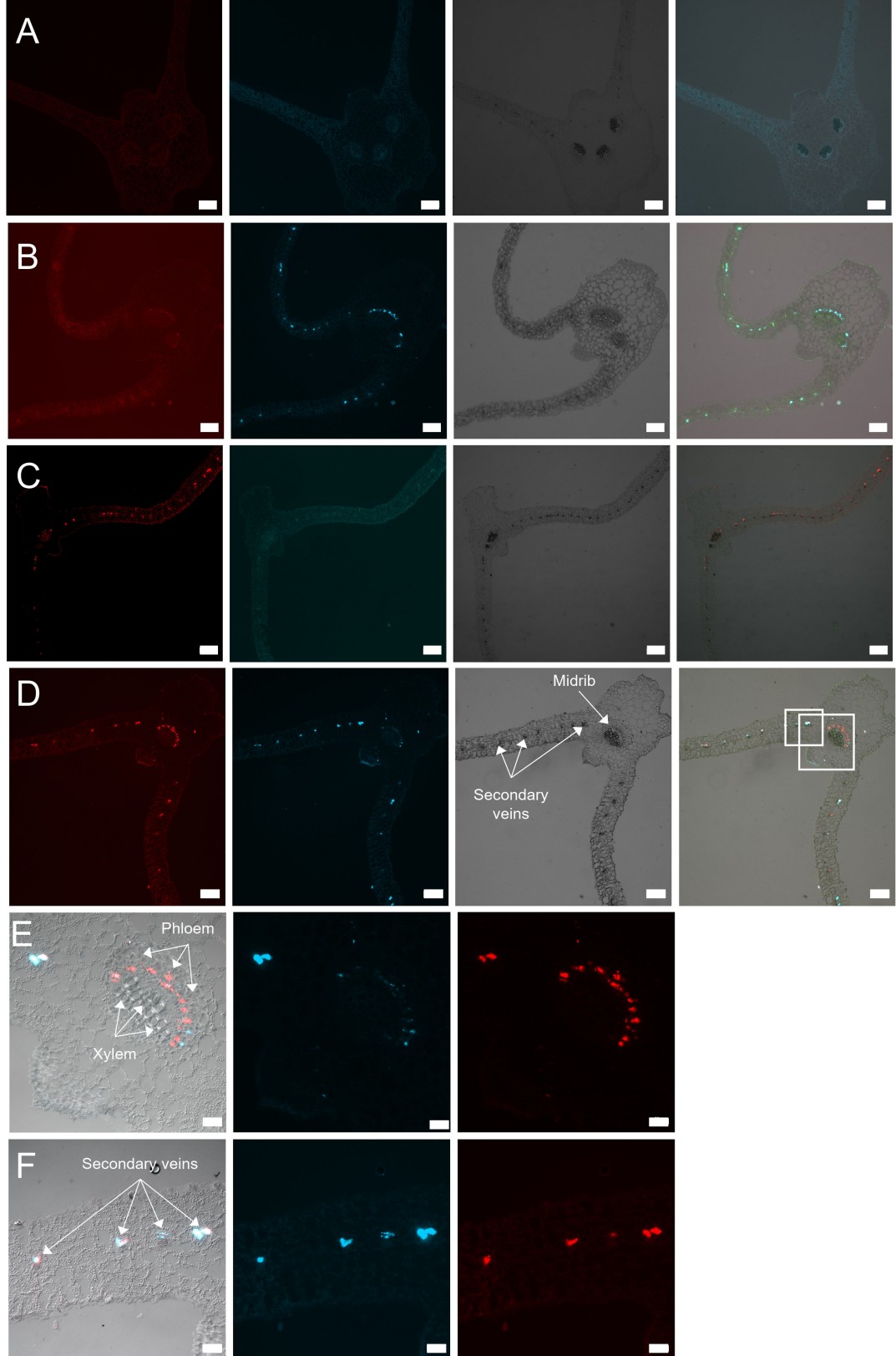

**FIG 4** Tissue distribution of BChV and BYV in leaves mono-infected with BChV and BYV or co-infected with both viruses. Leaves were processed 3 weeks after inoculation by SABER-FISH for detection of BChV and BYV. (A–D) Representative images showing transversal sections of (A) healthy, (B) BYV mono-infected, (C) BChV mono-infected, and (D) BChV/BYV co-infected leaves. The first column shows the BChV label (red); the second column shows the BYV label (turquoise); (Continued on next page)

Fig 4 (Continued)

the third column shows bright field acquisitions; and the last column presents merges. (E and F) Magnifications of the regions outlined in panel D showing virus distribution in (E) the midrib and (F) the leaf lamina. The first column shows image merges; the second column shows the BYV signal (red); and the last column shows the BChV label (red). Scale bars = 200 µm (A–D) and 50 µm (E and F).

spread uniformly in the cytoplasm of infected cells (Fig. 5B). Similar label patterns were also found in immunofluorescence experiments (Fig. S2), suggesting that the FISH label corresponded, at least partially, to virions.

In BYV/BChV co-infected plants, both viruses showed the same intracellular distribution as in mono-infected plants when the cells were infected by one virus only, i.e., pearl-like inclusions for BYV and homogenous cytoplasmic distribution for BChV (Fig. 6A and B). However, in co-infected cells, the BYV label was mostly dispersed in the cytoplasm and co-localized partially with the diffuse BChV label (Fig. 6C and D).

## DISCUSSION

Few studies have shown that viruses from various genera or families co-existing in multi-infection can modify vector transmission, and even fewer have addressed modifications at the cellular level. Additionally, to our knowledge, this is the first study examining the effect of co-infection by a polerovirus (BChV) and a closterovirus (BYV), both phloem-limited and sharing the same aphid vector, on transmission efficiency and other infection parameters. We found that co-infection of sugar beet with BYV and BChV reduced aphid transmission of BChV significantly, compared to transmission from mono-infected plants. No effect of the co-infection of sugar beet was observed on the transmission efficiency of BYV.

To find possible explanations for the reduction of BChV transmission from co-infected plants, we analyzed aphid feeding behavior, virus accumulation, and virus distribution in the leaves and the infected cells on mono- and co-infected plants.

Viral infections may impact aphid feeding behavior, which can have a direct impact on virus acquisition and transmission (12, 13). Expectations are that the effect of virus localization in the host on vector behavior is more important than its persistence in the vector (37). We confirm this here by showing that semi-persistent BYV and persistent BChV, both phloem-limited and acquired by aphids during phloem sap ingestion, induce similar host-mediated effects on vector behavior; i.e., we observed a neutral effect of BChV and a positive effect of BYV on phloem sap ingestion (Fig. 2).

The feeding behavior can be further modified in viral co-infections, but this has been hardly studied. One of the few published examples is that of raspberries co-infected with raspberry latent virus and raspberry leaf mottle virus. In this pathosystem, no changes in feeding behavior were observed, although aphid preferences for infected plants did change (16). Another example is the co-infection of melon plants with the potyvirus watermelon mosaic virus (WMV) and the crinivirus cucurbit yellow stunting disorder virus. There, WMV accumulation was 10 times lower in co-infected plants, but its transmission rate by aphids remained stable. EPG analysis showed that aphids on co-infected melons performed longer duration of ingestion (more specifically, pulses of the subphase Il-3) during intracellular test punctures in the epidermis and mesophyll,

TABLE 1  Number of cells infected by one or both viruses in sections of co-infected leaves

| Virus(es) observed | Exp. 1 | | | Exp. 2 | | |
|---|---|---|---|---|---|---|
| | Plant #1 | Plant #2 | Plant #3 | Plant #1 | Mean[a] | %[b] |
| BChV | 32[c] (33.0[d]) | 13 (20.6) | 3 (6.3) | 22 (34.4) | 17.5 ± 12.4 | 25.7 |
| BYV | 33 (34.0) | 19 (30.2) | 27 (56.3) | 21 (32.8) | 25 ± 6.3 | 36.8 |
| BYV + BChV | 32 (33.0) | 31 (49.2) | 18 (37.5) | 21 (32.8) | 25.5 ± 7.0 | 37.5 |

[a]Mean number of cells observed on 10 to 11 sections for four independent plants.
[b]Percentage of cells containing one or both viruses/total number of cells observed in the experiments.
[c]Number of cells observed on 10 to 11 sections of each plant showing a fluorescent signal for the indicated virus.
[d]Percentage (%) of cells labeled for the indicated virus/total number of labeled cells in 10 to 11 sections.

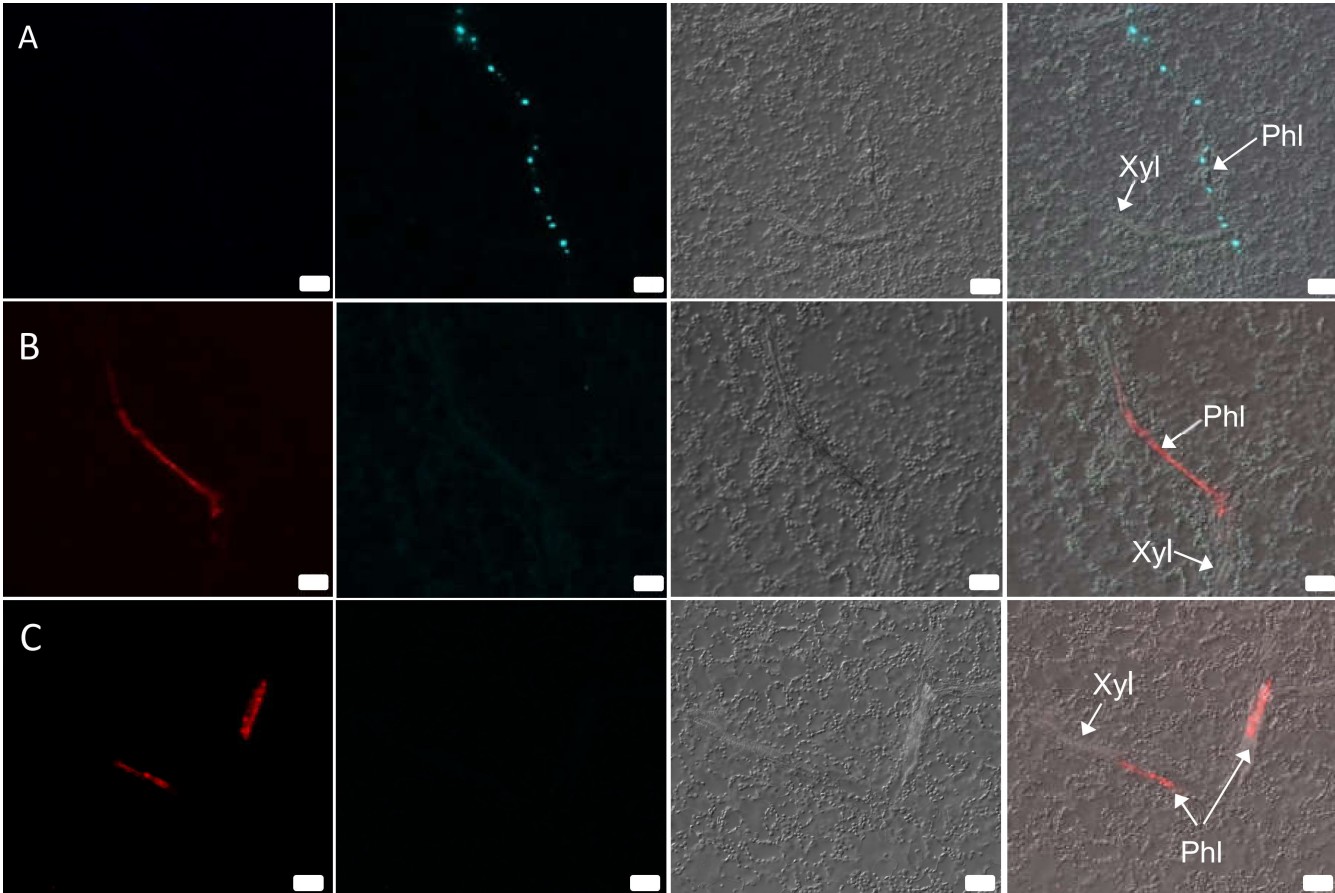

**FIG 5** Intracellular distribution of BYV and BChV in leaves of mono-infected plants. Leaves were processed 3 weeks after inoculation by SABER-FISH for detection of BYV and BChV. The images show longitudinal sections of (A) a leaf mono-infected with BYV and (B and C) a leaf mono-infected with BChV with virus label in the phloem in (B) probably a sieve tube and (C) probably in a sieve tube and in a companion cell (left and right arrows pointing from Phl, respectively). The first column represents the BChV label (red); the second column shows the BYV label (turquoise); the third column shows differential interference contrast images; and the last column presents image merges. Scale bars = 20 µm. Phl, phloem; Xyl, xylem.

which is the feeding behavior associated with the acquisition of non-persistently transmitted viruses like WMV. The authors argued that the changed feeding behavior might compensate for the low WMV accumulation in co-infected plants (38). Thus, aphid feeding behavior could be a determining factor that affects virus acquisition and explains the lowered transmission of BChV from co-infected plants. Since BChV should be acquired from the phloem sap, longer phloem sap ingestion should increase acquisition (39, 40), because BChV probably uses, like related poleroviruses (41), receptor-coupled endocytosis for intestinal uptake, a non-saturable mechanism because of replacement or recycling of used receptors. However, our results showed that despite aphids tending to ingest phloem sap for longer durations on co-infected plants than on mono-infected ones, transmission of BChV dropped. This indicates that the decreased transmission of BChV cannot be explained by aphid feeding behavior. Interestingly, for BYV, transmission from co-infected plants was not affected although aphids tended to feed less on co-infected plants than on mono-infected plants. We assume that under our experimental conditions, the 3 h of phloem sap ingestion observed for aphids feeding on co-infected plants is already sufficient to charge aphids maximally with BYV, since there is probably limited binding capacity in the aphid foregut and more specifically in the cibarium, where BYV could, like a related closterovirus, citrus tristeza virus, accumulate (42, 43). This is in line with the results by Jiménez and co-workers who demonstrated that maximum acquisition of BYV was obtained after 3 to 6 h of phloem sap ingestion, with

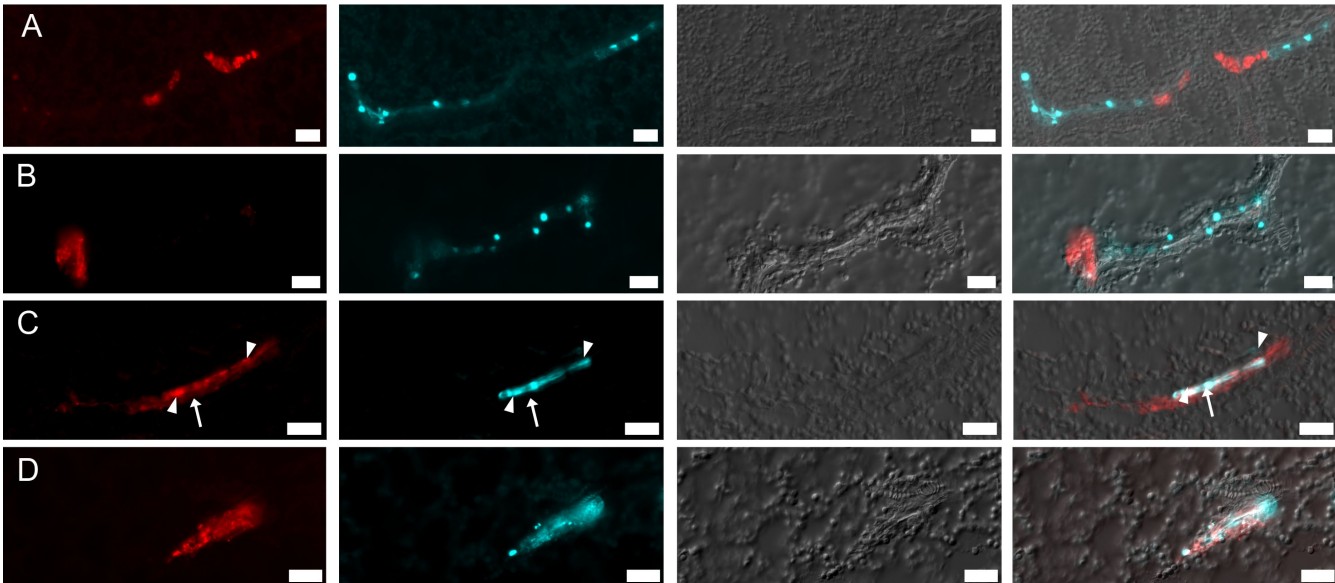

**FIG 6** Intracellular distribution of BYV and BChV in leaves of co-infected plants. Leaves were processed 3 weeks after inoculation by SABER-FISH for detection of BChV (red) and BYV (turquoise). The images show longitudinal sections of phloem with (A and B) mono-infected cells, (C) a cell mono-infected with BChV (arrow), and an adjacent cell co-infected with both viruses (arrowheads), and (D) a co-infected phloem cell. The first column presents the BChV label; the second column shows the BYV label; the third column shows differential interference contrast images; and the last column shows image merges. Scale bars = 20 µm.

no significant increases in transmission rates observed for longer sap ingestion periods (29). Taken together, we have no evidence that the altered aphid feeding behavior observed on co-infected sugar beets is the cause of the lower BChV transmission from these plants.

Lower or higher virus accumulation in co-infected compared to mono-infected plants may also directly affect transmission efficiency by modulating the amount of virus particles acquired by the vector. For instance, vector transmission efficiency of two criniviruses, ToCV and TICV, was linked to virus accumulation in mono- and co-infected tomatoes (4). However, an overall higher accumulation of one or both viruses in co-infected plants does not necessarily lead to increased vector transmission. Rather, the virus titer increase needs to occur in the aphid feeding sites (e.g., sieve tubes for polerovirus acquisition) (40). In this sense, a recent study showed that co-infection of tomato with a begomovirus and a crinivirus resulted in a synergistic effect on the accumulation of both viruses, but only increased the acquisition efficiency of the begomovirus (44). An increase in PLRV accumulation in *Nicotiana clevelandii* was observed when the plants were co-infected with the potato virus Y and the co-infection also induced phloem escape of PLRV (45). However, aphid transmission of PLRV and PVY was not addressed in this study. Our results showed that transmission of BChV drops when plants are co-infected with BYV, although BChV levels were not statistically different in mono- compared to co-infected plants. This indicates clearly that changes in aphid transmission rates of BChV are not linked to virus titer.

Another reason to explain altered transmission might be changes in the tissue tropism of viruses in plants that might modify virus accessibility to aphids. For instance, phloem escape of aphid-vectored PLRV in tobacco was reported in plants co-infected with pea enation mosaic virus 2 (PEMV-2 (46),) or PVY (45). The changed tissue specificity correlated with PLRV becoming mechanically transmissible in PLRV/PEMV-2 co-infections. In our observations, we did not find evidence of a phloem escape of BChV or BYV in co-infected plants, ruling out altered tissue tropism as the reason for reduced BChV transmission from co-infected plants. What we did observe in co-infected plants, though, was that BChV localized equally in the phloem of primary and secondary veins, while BYV was more present in those of secondary veins. However, since both viruses are acquired

preferentially from sieve tube sap, it is unlikely that the tissue relocalization impacted virus acquisition greatly.

We found that in co-infected phloem cells (representing about 40% of all infected cells), the intracellular distribution of BYV was modified. BYV was present as spherical, cytoplasmic inclusions aligning in chains in cells of mono-infected plants. BChV was spread uniformly in the cytoplasm in cells of mono-infected plants. In co-infected plants, both viruses displayed the same intracellular localization when cells were infected by only one virus. This was different in co-infected cells. While BChV maintained its diffuse distribution, the typical BYV inclusions disappeared and BYV label was diffuse in the cytoplasm and co-localized partially with BChV. We take this as evidence that the two viruses interact directly or indirectly with each other. Since no other notable differences were detected in co-infected vs mono-infected plants, we propose that virus-virus interactions in co-infected cells might have caused the drop in BChV transmission. However, the mechanisms of interaction between BChV and BYV remain unknown at this stage. Because BChV should be acquired predominantly from the phloem sap and only to a small extent from phloem cells (40), one hypothesis is that co-infection limits BChV release from companion cells into the sieve tubes (Fig. 7). This would result in a lower BChV accessibility to aphids, compared to BChV mono-infected plants. This could be achieved by alterations of the intracellular virus trafficking or modifications of the cell wall or plasmodesmata. Several BYV and BChV proteins are known to be involved in virus movement and interaction with plasmodesmata and are therefore good candidates for further studies (33, 47).

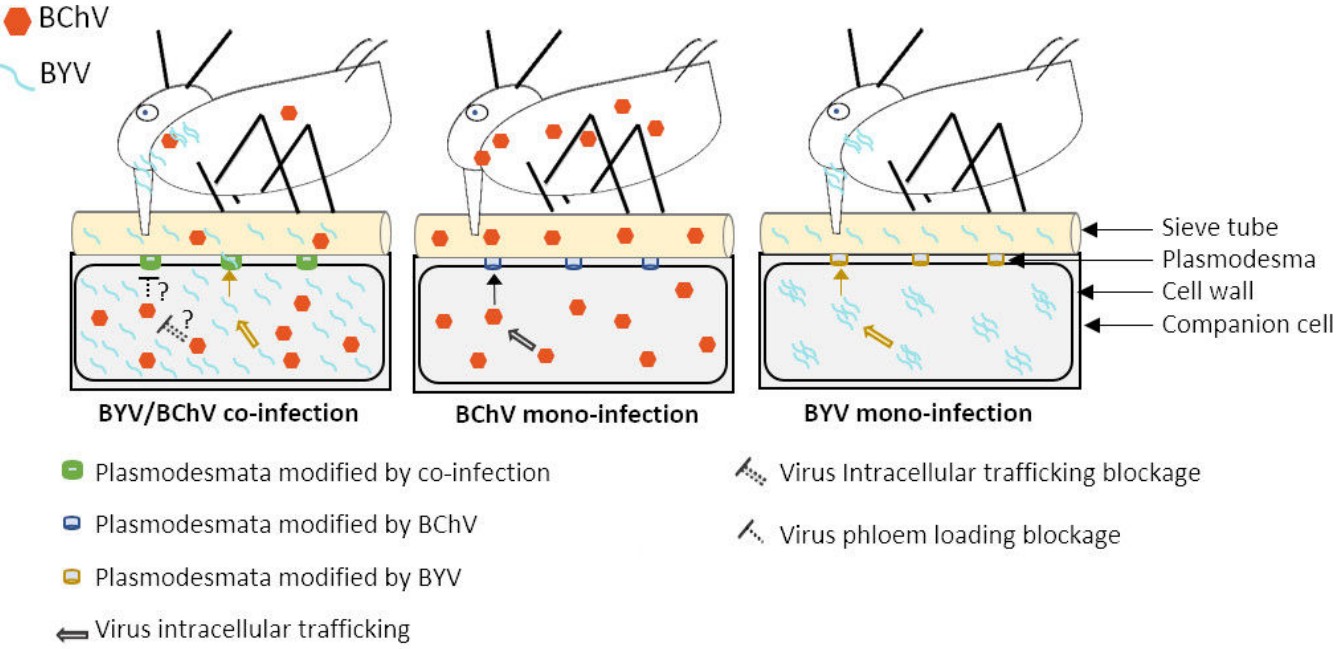

FIG 7 Model for explaining the reduction of BChV transmission from BChV/BYV co-infected plants. In mono-infected plants (middle and right cells), BChV and BYV particles are dispatched from infected companion cells through the plasmodesmata into the sieve tube lumen and acquired by aphids ingesting phloem sap from the sieve tubes. In co-infected plants, roughly one-third of the infected phloem cells are co-infected. Co-infection does not affect virus accumulation measurably, and the two viruses might even accumulate to similar levels in mono-infected and co-infected cells, as presented in the schema (left cell). BYV and BChV interact in co-infected cells directly or indirectly. This inhibits the passage of BChV toward and/or through plasmodesmata and BChV release into the phloem sap, whereas phloem sap loading with BYV remains unchanged. Consequently, the BChV load in the sieve tube is significantly lower, while that of BYV is not changed, and aphids acquire and transmit less BChV from the sieve tubes of co-infected plants than from those of mono-infected plants. Two scenarios are thinkable that are not mutually exclusive: co-infected phloem cells inhibit BChV release totally and the BChV particles in the phloem sap origin from BChV mono-infected cells, or BYV infection interferes with phloem loading from all companion cells. In the former case, BYV operates only in co-infected cells; the latter case requires the movement of a signal or factor from BYV-infected cells to uninfected and BChV-infected cells.

Our results show that the percentage of co-infected cells and those mono-infected with BYV or BChV were in the same range (about 30%) in co-infection. This suggests that co-infection is a random event and that the viruses do not exclude each other. Further, there was no alteration in the accumulation of either virus, indicating that there is no competition for plant resources and that they do not interfere with each other's replication. This is indicative of neutralism. This assumption is also reinforced by the fact that BYV/BChV co-infection, compared to infection with BYV alone, does not have a more severe impact on sugar beet root mass (https://www.itbfr.org/publications/fiches-bioagresseurs/les-jaunisses-virales-et-leurs-pucerons-vecteurs/, accessed on 4 April 2024). Note that the same sugar beet cultivar was used in both studies.

In conclusion, our results shed some light on the mode of interaction between two unrelated viruses, a polerovirus and a closterovirus, a combination that has been studied before (20, 21) but not in such depth. Except for the antagonism observed for BChV transmission by aphids, the outcome of the sugar beet co-infection with BChV and BYV is fairly neutral for all other parameters analyzed. The reasons for BChV decreased aphid transmission by aphid are unknown, but the subcellular localization of both viruses in co-infected plants strongly suggests interactions between them that are worth exploring. Finally, the alteration of aphid transmission of BChV from co-infected plants might have consequences at the epidemiological level. For example, it might explain partially the huge changes in incidences of the two viruses from 1 year to another (20).

## MATERIALS AND METHODS

### Plants, viruses, and aphids

The virus-susceptible sugar beet variety "Auckland" was used in this study. Seeds were kindly provided by the seed company SESVanderHave (Tienen, Belgium) via the Technical Institute of Sugar Beet (Paris, France). One week after germination, plantlets were transplanted into individual pots and grown in a climate chamber at 22°C–25°C with a day/night cycle of 16 h of light and 8 h of darkness.

The *Myzus persicae* biotype WMp2 (local code NL) was used in this study (48). Aphids were maintained on Chinese cabbage (*Brassica rapa* subsp. *Pekinensis* var. Granaat) in a growth chamber under controlled conditions at $20 \pm 1°C$ with a day/night cycle of 16 h of light and 8 h of darkness.

The BYV_BBRO_UK isolate used in this study was a kind gift from Prof. Mark Varrelmann of the Institut für Zuckerrübenforschung (Göttingen, Germany) but was originally sampled in the UK (49). The isolate was maintained on sugar beet plants and propagated by aphid transmission.

We used the BChV-2a isolate collected in East England (UK) with the GenBank accession number AF352024 (50). BChV particles, prepared as described in (51), were maintained at −80°C before being used as a virus source in an artificial medium for aphid acquisition (52).

For all experiments, we used the systemically infected third upper leaf at $21 \pm 2$ dpi.

### Virus detection by DAS-ELISA

BChV and BYV were detected in leaves of sugar beets at 21 dpi by double-antibody sandwich enzyme-linked immunosorbent assay [DAS-ELISA (53)] with turnip yellows virus antibodies that recognize BChV as well (our observation) and beet yellows virus-specific antibodies, respectively (Loewe Biochemica, Sauerlach, Germany). Samples were considered infected when the $OD_{405 \text{ nm}}$ was twice the mean value of non-infected control plants (three technical replicates) plus three times the standard deviation.

## Generation of mono- and co-infected source plants for subsequent experiments

Non-viruliferous aphids were allowed a 24-h acquisition access period on BYV-infected plants or on purified BChV particles. Then they were transferred to 2-week-old healthy sugar beet plants for a 3-day inoculation access period. About 10 aphids of each condition were placed simultaneously on a plant for mono- and co-infection. Aphids were then removed manually. Inoculated and healthy, non-inoculated control plants were maintained in a growth chamber under controlled conditions (22°C–25°C and 16/8 h light/dark photoperiod) for 3 weeks before being analyzed by DAS-ELISA to verify infection.

## Aphid transmission experiments

A 24-h acquisition access period on detached source leaves (three leaves from three independent infected source plants that were placed on 1% agarose in Petri dishes) was followed by a 72-h inoculation access period using 15 test plants per condition and three aphids per test plant. Plants were treated 3 days later with pirimicarb insecticide to kill aphids. After 3 weeks' growth in a climate chamber as indicated above, plants were analyzed by DAS-ELISA, and the transmission efficiency of each virus from co-infected plants was compared to that of transmission of each virus from mono-infected plants.

## Localization of viruses by SABER-FISH

For virus localization, we used SABER-FISH, a multiplexable nucleic acid-based detection method. It includes a signal amplification step enabling highly sensitive target revelation. We used the protocol that was originally developed for frozen retinal tissue as described by Kishi et al. (54) with some adaptations to optimize the technique for plant tissues. For the detection of BYV and BChV, we designed 25 (BYV) and 16 (BChV) primers with OligoMiner (55). A TTT and a concatemer sequence were added to the 3′-end of each primer (concatemer 27 for BChV and 26 for BYV, see Table S1).

Concatemers were added to the 3′-end of the probes by *in vitro* primer exchange reaction (PER) to obtain probes of 300–400 nucleotide lengths (54). The product of the PER reaction is a probe consisting of two parts: a specific sequence that recognizes the target and the concatemer part that enables target detection and amplification of the signal by binding a concatemer-specific fluorescent imager oligonucleotide. After concatemer extension, probes were purified and concentrated using MinElute PCR purification columns (Qiagen, Courtaboeuf, France). Probe concentrations after purification were determined with a Nanodrop 2000 spectrometer (Thermo Fisher Scientific, Illkirch-Grafenstedt, France) using ssDNA setting. A final concentration of 800 ng/μL of the probe was used for hybridization.

To prepare paraffin-embedded tissues, rectangular (10 mm long and 5 mm wide) samples from the third upper leaf and encompassing the midrib were excised with a razor blade and immersed immediately in 10 mL ice-cold 4% paraformaldehyde solution in PBS. Samples were fixed by applying a vacuum (~720 mm Hg) overnight at 4°C. After dehydration in a graded series of ethanol/water, followed by a graded series of ethanol/xylene and xylene/paraffin, the samples were embedded in paraffin (Paraplast, Leica, Nanterre, France). Sections of 10 μm were cut using a rotary microtome and immobilized on Superfrost + slides. After de-waxing and rehydration, the fixed tissue samples were processed for SABER-FISH as described (54). Briefly, slides were first prehybridized in pre-hybridization buffer, then hybridized with all PER probes simultaneously at 43°C. This hybridization step was followed by a short second hybridization step at 37°C to bind the fluorescent imagers (with ATTO-565 or ATTO-647N dyes conjugated to their 5′-ends) to the probes. Slides were mounted in Fluoroshield medium (Sigma, Saint-Quentin-Fallavier, France) and observed with a Zeiss Axio Imager M2 microscope equipped with a Hamamatsu Orca-Flash 4.0LT black-and-white camera (Lordil, Lay-Saint-Christophe, France). Images were acquired with a ×5, ×20, or ×40 objective and with

Zeiss 43 HE or AHF Cy5 ET bandpass filter sets for epifluorescence acquisition in the red or far-red channels, respectively, or with brightfield or DIC settings for transmission light acquisition. Images were processed with ZEN 2.5 or Image J 1.54 software. Final figures were assembled with Microsoft PowerPoint or LibreOffice Impress.

## Relative virus quantification by one-step TaqMan RT-qPCR

Gene-specific primer pairs and TaqMan probes for BYV and BChV were designed in collaboration with GEVES (Angers, France) using the Primer Express version 3.0.1 primer design tool (Thermo Fisher Scientific). Primer sets were designed to amplify between 75 and 200 nucleotides of the BYV and BChV genomes (Table 2). The specificity of the primer pairs was controlled. Each primer pair was evaluated by a standard curve with six dilutions and three technical replicates of BChV- or BYV-infected plants. Efficiency rates ($E$) of 95% (for BYV) and 94% (for BChV) were obtained with $R^2 = 0.99$ for both viruses.

To select reference genes, we used a list of stably expressed plant genes previously established on *Arabidopsis thaliana* infected by several viruses (56). Orthologous genes were identified in *Beta vulgaris* using the Kyoto Encyclopedia of Genes and Genomes database, and primers compatible with TaqMan and SYBR Green detection were designed using the IDT Primer Request tool available online (https://eu.idtdna.com/pages/tools/primerquest). We tested the suitability and stability of the reference genes by SYBR Green PCR. Of the nine tested reference genes, five were selected with an amplification efficiency ranging from 90% to 99% and with $R^2$ correlation values for the curves ranging from 0.98 to 0.99 (Table S2). The expression stability of candidate genes was then verified on five biological replicates per treatment (mono- or co-infected sugar beets) at 21 dpi. For each biological replicate, three technical replicates were analyzed. Two hundred-nanogram RNA was used to prepare complementary DNA (cDNA) using M-MLV reverse transcriptase kit (Promega, Charbonnières-les-Bains, France) and oligo(dT) nucleotides. qPCR for reference gene evaluation was performed using Universal SYBR Green Supermix (Bio-Rad, Marnes-la-Coquette, France) with gene-specific primers (20 µM) in a CFX96 Touch Deep Well Real-Time PCR detection system (Bio-Rad). Two microliters of initial cDNA was used as template in 10-µL qPCR reaction to evaluate the expression of reference genes. The cycling conditions were 98°C for 2 min, followed by 40 cycles of 15 s at 98°C, 20 s at 65°C, with an increment ramp of 0.5°C/s. Data from these experiments were entered into geNorm (57), Bestkeeper (58), and Normfinder (59) programs for gene stability validation (60, 61).

Relative virus accumulation was measured on five biological replicates using AgPath-ID One-step RT-PCR Kit for TaqMan chemistry (Applied Biosystems by Thermo Fisher Scientific). The third upper systemically infected leaf from each source plant was collected at 21 dpi. The whole leaf was homogenized in liquid nitrogen. One hundred milligrams of ground tissue was used for total RNA extraction using NucleoSpin RNA Plant Kit (Macherey-Nagel, Hoerdt, France). After spectrometric quantification with a Nanodrop 2000, 30 ng of RNA was used in a 23-µL reaction volume. The cycling

**TABLE 2** Primer and probe sequences used for virus quantification by RT-qPCR

| Virus/sugar beet gene | Sequence (5′–3′) | Primer orientation and probe |
|---|---|---|
| BYV | 5′-TAC TGT TCC AAA CCA GGT CCT TG-3′ | Forward |
| | 5′-ROX-TTG CTT CTT TTT CAA CTC CAC CAC CCT GT-BHQ-2-3′ | Probe |
| | 5′-GTG CAA CGC AGT TCG AAA CTA A-3′ | Reverse |
| BChV | 5′-GGG ACC ATG GCA CCA TCT T-3′ | Forward |
| | 5′-VIC-TCC CTT ACC ACC GGA TAT TAC CCA ACT CCT- BHQ-1-3′ | Probe |
| | 5′-GTA ATC TGA CAG CTT TTT CTG AAG AGG-3′ | Reverse |
| Tub[a] | 5′-GGCTTTCTTGCATTGGTACAC-3′ | Forward |
| | 5′-FAM-CGAGGCTGAGAGCAACATGAACGA-BHQ-1-3′ | Probe |
| | 5′-CATCCTGGTACTGCTGGTATTC-3′ | Reverse |

[a]Tubulin.

conditions were 95°C for 10 min, followed by 40 cycles of 15 s at 95°C and 1 min at 65°C, using a CFX384 Touch Real-Time PCR Detection System (Bio-Rad). The tubulin gene was used as a reference gene and the relative accumulation was determined using the $2^{-DDCt}$ method (62).

## Aphid feeding behavior

We used the EPG system (Giga-8 DC-EPG amplifier; EPG Systems, Wageningen, Nether-lands) as described by Tjallingii (63) to investigate the feeding behavior of *M. persicae*. We created electrical circuits that each included one aphid and one plant by tethering a thin gold wire (12.5-μm diameter and 2 cm long) on the insect's dorsum using conductive silver glue (EPG Systems). To facilitate the tethering process, aphids were immobilized at the edge of a pipette tip with negative pressure generated with a vacuum pump (model N86KN.18; KNF Neuberger, Freiburg, Germany). Eight apterous adult aphids (sampled from the rearing plant) were connected to the Giga-8 DC-EPG amplifier (EPG Systems), and each one was placed on the adaxial side of the third upper leaf of an individual plant at 18–23 dpi. The circuit was closed by inserting a copper electrode into the plant substrate. The feeding behavior was recorded for 8 h at a temperature of 20 ± 1°C and constant light inside a Faraday cage. Each plant and aphid were used only once. Plants from all conditions (non-inoculated, infected by BChV or BYV or co-infected) were used on the same day to run EPGs to avoid bias due to varying external conditions. Acquisition of the EPG waveforms was carried out with PROBE version 3.5 software (EPG Systems). This software was also used to identify the different waveforms correlated with the phases of *M. persicae* feeding behavior, as described (64). The Excel workbook developed by Sarria et al. (65) was used to calculate the parameters of EPG data. For a given behavior, the parameters of occurrence and total duration were measured and used to analyze the feeding behavior. We decided to select the following parameters for analysis because they are relevant for the acquisition of phloem-limited viruses by aphids, i.e., the total durations and occurrences of stylet penetration, pathway phases, extracellular salivation phase, phloem sap ingestion phases, and the time needed by the aphid to perform the first phloem phase.

## Statistical analyses

Statistical analyses were made with R version 4.3.2 (https://www.r-project.org/).

Differences in virus accumulation between mono- and co-infections were determined using a student test.

We used generalized linear models (GLMER, package: "lme4") with a likelihood ratio and $\chi^2$ test to assess whether co-infection affected virus transmission. Data on transmission rate were analyzed following a binomial error distribution, and "session" was treated as a random factor.

We used generalized linear model (GLM) with the likelihood ratio and the $\chi^2$ test to assess whether mono or co-infection affected aphid feeding behavior. As feeding duration parameters were not normally distributed, we used GLM using a gamma (link = "inverse") distribution, and parameters related to frequency of penetration were modeled using GLM with Poisson (link = "identity") distribution. The parameter "time to first phloem phase" was modeled using the Cox proportional hazards model, and we treated cases where the given event did not occur as censored. The assumption of the validity of proportional hazards was checked using the functions "coxph" and "cox.zph", respectively (R packages: "survival" and "RVAideMemoire").

The fit of all generalized linear models was controlled by inspecting residuals and QQ plots. When a significant effect was detected, to test for differences between treatments, a pairwise comparison using estimated marginal means (package R: "emmeans") (*P* value adjustment with Tukey method) at the 0.05 significance level was used.

## ACKNOWLEDGMENTS

Plants were produced by the experimental unit of INRAE Grand Est-Colmar (UEAV). We thank Samia Djennane for help with RT-qPCR, Amandine Velt for help with probe design, and Danaé Brun for help with virus identification.

This work was financed by the PNRI project ProViBe awarded to V.B. and the ANR PRCI project ManiVir (ANR-23-CE20-0049-01) awarded to M.D. S.K. received additional funding from INRAE SPE department. The funders had no role in study design, data collection and analysis, decision to publish, or preparation of the manuscript.

S.K.: conceptualization, methodology, validation, formal analysis, investigation, writing (original draft preparation, review, and editing), and visualization; Q.C.: methodology, formal analysis, and writing (review and editing); C.V.: resources; V.B.: conceptualization, methodology, investigation, writing (original draft preparation, review, and editing), supervision, project administration, and funding acquisition; M.D.: conceptualization, methodology, investigation, writing (original draft preparation, review, and editing), visualization, supervision, project administration, and funding acquisition.

## AUTHOR AFFILIATIONS

[1]SVQV, UMR 1131, INRAE Centre Grand Est, Colmar, France
[2]Université Strasbourg, Strasbourg, France

## AUTHOR ORCIDs

Véronique Brault  http://orcid.org/0009-0007-9524-6693
Martin Drucker  http://orcid.org/0000-0002-9765-1189

## FUNDING

| Funder | Grant(s) | Author(s) |
|---|---|---|
| PNRI | Provibe | Véronique Brault |
| Agence Nationale de la Recherche (ANR) | ManiVir ANR-23-CE20-0049-01) | Martin Drucker |
| SPE | Scholarship | Souheyla Khechmar |

## AUTHOR CONTRIBUTIONS

Souheyla Khechmar, Conceptualization, Formal analysis, Investigation, Methodology, Validation, Visualization, Writing – original draft, Writing – review and editing | Quentin Chesnais, Formal analysis, Methodology, Writing – review and editing | Claire Villeroy, Resources | Véronique Brault, Conceptualization, Funding acquisition, Investigation, Methodology, Project administration, Supervision, Writing – original draft, Writing – review and editing | Martin Drucker, Conceptualization, Funding acquisition, Investigation, Methodology, Project administration, Supervision, Visualization, Writing – original draft, Writing – review and editing

## DATA AVAILABILITY

All relevant data are within the article and its supporting information files.

## ADDITIONAL FILES

The following material is available online.

### Supplemental Material

**Data S1 (Spectrum01115-24-s0001.xlsx).** Transmission experiments.
**Data S2 (Spectrum01115-24-s0002.xlsx).** EPG parameters.

**Data S3 (Spectrum01115-24-s0003.xlsx).** RT-qPCR virus accumulation.
**Data S4 (Spectrum01115-24-s0004.xlsx).** Co-infection cell counts.
**Supplemental material (Spectrum01115-24-s0005.pdf).** Fig. S1 and S2; Tables S1 and S2.

## Open Peer Review

**PEER REVIEW HISTORY (review-history.pdf).** An accounting of the reviewer comments and feedback.

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
