## [Reviewer comments · Microbiology Spectrum]

Microbiology Spectrum

Interplay between a polerovirus and a closterovirus decreases aphid transmission of the polerovirus

Souheyla Khechmar, Quentin Chesnais, Claire Villeroy, Véronique Brault, and Martin Drucker

Corresponding Author(s): Martin Drucker, SVQV INRAE Centre Grand Est Colmar

Review Timeline:

Submission Date:	May 3, 2024
Editorial Decision:	July 17, 2024
Revision Received:	July 31, 2024
Accepted:	August 11, 2024

Editor: Clinton Jones

Reviewer(s): Disclosure of reviewer identity is with reference to reviewer comments included in decision letter(s). The following individuals involved in review of your submission have agreed to reveal their identity: Juan José López-Moya (Reviewer #3)

Transaction Report:

DOI: <https://doi.org/10.1128/spectrum.01115-24>

Re: Spectrum01115-24 (Interplay between a poliovirus and a closterovirus decreases aphid transmission of the poliovirus)

Dear Dr. Martin Drucker:

Thank you for the privilege of reviewing your work. Below you will find my comments, instructions from the Spectrum editorial office, and the reviewer comments. There are a few minor modifications that were raised by the second Reviewer. When these concerns are addressed, this manuscript will be accepted.

Revision Guidelines

Sincerely,
Clinton Jones
Editor
Microbiology Spectrum

Reviewer #2 (Comments for the Author):

In my opinion, the work is relevant and provides novel and interesting information about the virus-virus interactions under a more often mixed-infection scenario in agriculture. The focus of the work is interesting, with the objective of the work also well described and the methods to achieve quite appropriate. I have some comments to the manuscript for improvement and

clarification.

Reviewer #3 (Comments for the Author):

The manuscript by Khechmar and collaborators describes the effect on transmission efficiency of plant viruses by insect vectors, comparing single and mixed infections. The pathosystem under study is an interesting one, with two aphid-transmitted and phloem-restricted viruses, the polerovirus BChV and the closterovirus BYV, infecting simultaneously a common susceptible host. A main difference between the two viruses for the purpose of the work is the transmission mode: persistent circulative, and semipersistent, respectively for the polerovirus and the closterovirus. Approaching to the study of mixed viral infections in plants represents a challenge, and consequently there are few in depth analysis like the present paper. Furthermore, the transmission process by insects add another layer of complexity, since it involves managing vectors in the experimental set up. All together, the effort of the authors must be acknowledged, since the work brings novel insights to a difficult topic.

The manuscript is well written, the rationale of the experiments is clearly described, and the results are presented in a comprehensive manner. In order to find explanation(s) to the observed effect on transmission % for BChV, authors measure vector behavior parameters using EPG, virus accumulation, and distribution of the two viruses in the plants, both in tissues and intracellularly, finding some differences in mixed infections compared to single infections regarding the redistribution of BYV. They propose that this redistribution could explain the reduced transmissibility through virus-virus interactions, and provide a model to illustrate the case.

The data provided is solid, and only a certain speculation is leaved for the mechanism of virus-virus interaction. Considering that this interaction might be indirect/more complex than a testable binding of the two virions, and it might involve other viral products, the discussion of the case (l. 475-491) is quite fair.

Minor points and other aspects to be addressed by authors in a revised version:

- L. 41. Authors define "co-infection, multi-infection or mixed infection" as simultaneous presence of more than one pathogen in a given host. However, later in the same paragraph (l. 47), they differentiate correctly co-infection and super-infection as the results of the simultaneous or sequential arrival of the two pathogens, respectively. I agree with this idea to separate both concepts, therefore it might be better not to use co-infection in l. 41. Later it will be a good idea to state at the beginning of the results section that all the experiments were performed with co-infected plants (as described in methods, l. 157-163).
- L. 321. Correct misspelling of the statistical method employed: Tukey HSD
- L. 502. Add a short statement here to clarify that the case of study was not unknown to science, although it can be said that it was not previously addressed in the same depth including the transmission perspective.

- In Figure S1 is not necessary to plot the "expected" transmission rate, which is calculated theoretically from the data already presented in figure 1. Fig S1 can be better replaced by a table, or to provide the experimental data in the main text (l. 292).

Interplay between a polerovirus and a closterovirus decreases aphid transmission of the polerovirus

In my opinion, the work is relevant and provides novel and interesting information about the virus-virus interactions under a more often mixed-infection scenario in agriculture. The focus of the work is interesting, with the objective of the work also well described and the methods to achieve quite appropriate. I have some comments to the manuscript for improvement and clarification.

Methods

Line 158: Here it is indicated that aphid acquired BChV from purified virions rather than a infected host plant? Is there any reason for this procedure?

Also indicate here how the single-infected plants were generate (not only the double infected)

Lines 158: Specify the AAP on sugar beet for later generation of the source plants

Line 160: the third leave is already mentioned in the first methods section

Lines 165: ‘Aphids acquired viruses for 24 h from detached source leaves (three leaves from three independent’

In this case it is only referred to BYV right? Authors mention previously hat BChV was acquired from purified virions. It is not clear. Please, clarify this.

You should also indicate here where the test plants were kept until virus detection with ELISA.

Line 166: use always IAP when referring to virus inoculation Access period or AAP for acquisition access period

Line 188: what kind of leaf and specific area of the leaf was used for the SABER-FISH studies? And also, describe a bit how this section was obtained with more details

Lines 244: aphid feeding behavior section:

Here, authors should clarify some details regarding the methodology for the EPGs. First, you should indicate here the type of plants (time after infection, etc) used for these experiments (even though they are already indicated in results). Also, even it could be obvious; authors should state here that plants from different treatments were used the same day to run EPGs to avoid bias of external climatic conditions. Also, some small but important details are the abaxial or adaxial side, age of aphids, adults or nymphs etc.

Also, even though there is a section about how these co-infected plants were generated (**‘Establishing mixed infection in source plants’**), the section is confusing and should be clarified and extended. How single-infected plants were generated, how the mock plants were generated (using non-viruliferous aphids?) This is really important as these co-infected plants are one of the pivot material of the manuscript for both EPG and virus detection in the plant tissues. For example, it is not clear whether both viruses were inoculated at once or in subsequent inoculations, or if same aphid acquired

sequentially both viruses and then inoculated them in the test plants. Authors should provide a detailed description of how these plants were generated, number of aphids used, age of the healthy plants, age of the plants when used in the experiments, etc.

Results/discussion

Lines 213: **‘However, aphids spent significantly more time (~ one extra hour) ingesting phloem sap (E2 phase) on BYV mono-infected than on healthy or BChV-infected plants.’**

For me, that is a quite interesting result. One could suggest that a persistently infected plant could maybe promote sap phloem ingestion. However, this is only observed for the semipersistent virus (at least 1.5 h, which is a lot in a 8h recording). Even though that is not the focus of the work, I think it would be a quite small discussion about this fact. In fact, the feeding times observed for E5 correlate quite well with the transmission rates obtained. Even though more time was spent in E2 in the co-infected plants than in the single-BChV-infected plants, this even reduced the acquisition from the double-infected plants. This was not observed for BYV (the other way around, but this did not affected transmission for the semipersistent). I think authors should go deeper on this interesting fact in the discussion and supporting on previous works.

The stage in which the virus source plants were used for the transmission experiments was the same as that used for the EPG experiments?

Also, some variables are missing significance. Please, add it to all bars.

Lines 440: **We assume that under our experimental conditions, the 3 h sap ingestion observed for aphids feeding on co-infected plants is sufficient to charge aphids maximally with BYV and that for this reason.**

That is true for BYV, but why the same effect is not observed for mixed- infected don the transmission of BChV

Lines 494: **At this stage, the exact mechanism of interaction between BChV and BYV is unknown. Because BChV should be acquired predominantly from the phloem sap and only to a small extent from phloem cells (Prado & Tjallingii, 1994), we propose that co-infection limits BChV release from companion cells into the sieve tubes (Fig. 7).**

Authors could not support their results based on EPG nor virus titer or distribution and suggested that the decrease in the BChV transmission in the co-infected cells could be

explained due to interaction between viruses with the cell.

In my opinion, this assumption is not well supported by the results obtained in the manuscript. It could a nice way to find an explanation to the decrease in transmission observed, however, there are no clear results supporting that hypothesis. Probably authors should be a bit cautious about this assumption, just referring it as a likely fact explain the results, among other possible reasons behind, or dealing with it in a more extensive paragraph if associating that hypothesis with published bibliography on the topic.

We would like to thank the reviewers for evaluating our work and the positive and constructive criticism. Please find here our responses (in black) to your comments (in red). Line numbering refers to the unrevised submitted version used by the reviewers.

Response to reviewer 2:

Thanks for evaluating our work.

Methods

Line 158: Here it is indicated that aphid acquired BChV from purified virions rather than a infected host plant? Is there any reason for this procedure? Also indicate here how the single-infected plants were generate (not only the double infected)

We use purified BChV as virus source in routine. This is to avoid plant contamination with viruliferous aphids in the lab. Since we have not so far succeeded to obtain purified BYV particles that can be used in an artificial medium, infected BYV sugar beets were used as virus source. We modified the title and the text for description of source plants (please see below our response to (**Establishing mixed infection in source plants**)).

Lines 158: Specify the AAP on sugar beet for later generation of the source plants

We added AAP to the text Line 158. It reads now “*were allowed a 24 h acquisition access period on BYV-infected plants or on purified BChV particles.*” AAP was the same for the generation of source plants and for transmission experiments.

Line 160: the third leave is already mentioned in the first methods section

Thanks for noting the repetition. We removed the sentence Lines 160-161 (“*The third upper leaves (just below the two youngest leaves) were always used as the virus sources.*”).

Lines 165: ‘*Aphids acquired viruses for 24 h from detached source leaves (three leaves from three independent*’ In this case it is only referred to BYV right? Authors mention previously hat BChV was acquired from purified virions. It is not clear. Please, clarify this.

You should also indicate here where the test plants were kept until virus detection with ELISA.

To generate source plants for subsequent transmission experiments, we used BYV-infected sugar beet and purified BChV virions. For aphid transmission experiments, we used plants mono- and co-infected with BChV and/or BYV. To better distinguish between the two procedures, we modified the title Line 157. It reads now “*Generation of mono- and co-infected source plants for subsequent experiments*” To indicate the growth conditions of the test plants, we added to the text Line 168 “*After 3 weeks growth in a climate chamber as indicated above,...*”.

Line 166: use always IAP when referring to virus inoculation Access period or AAP for acquisition access period

Thank you for the suggestion, but we prefer to keep the present wording to avoid the use of too many abbreviations. But we adapted the terminology. The “Aphid transmission experiments” reads now “*A 24 h acquisition access period on detached source leaves (three leaves from three independent infected source plants that were placed on 1% agarose in Petri dishes) was followed by a 72 h inoculation access period using 15 test plants per condition and three aphids per test plant. Plants were treated three days later with pirimicarb insecticide to kill aphids. After 3 weeks growth in a climate chamber as indicated above, plants were analyzed by DAS-ELISA, and the transmission efficiency of each virus from co-infected plants was compared to that of transmission of each virus from mono-infected plants.*”

Line 188: what kind of leaf and specific area of the leaf was used for the SABER-FISH studies? And also, describe a bit how this section was obtained with more details

We describe sample preparation now in more detail: *“To prepare paraffin-embedded tissues, rectangular (10 mm long and 5 mm wide) samples from the third upper leaf and encompassing the midrib were excised with a razor blade and immersed immediately in 10 ml ice-cold 4% paraformaldehyde solution in PBS.”*.

Here, authors should clarify some details regarding the methodology for the EPGs. First, you should indicate here the type of plants (time after infection, etc) used for these experiments (even though they are already indicated in results). Also, even it could be obvious; authors should state here that plants from different treatments were used the same day to run EPGs to avoid bias of external climatic conditions. Also, some small but important details are the abaxial or adaxial side, age of aphids, adults or nymphs etc. Also, even though there is a section about how these co-infected plants were generated

We added the required information to the text. We used adult aphids (*“adult”* and *“sampled from the rearing plant”* added to Line 251 for more precision), but we cannot tell the precisely the age of the aphids, because this would have required daily synchronization of aphids, which is very demanding. As requested by the reviewer, we added to Line 255 information on how EPG was performed *“Plants from all conditions (non-inoculated, infected by BChV or BYV or co-infected) were used the same day to run EPGs to avoid bias due to varying external conditions.”*.

(**‘Establishing mixed infection in source plants’**), the section is confusing and should be clarified and extended. How single-infected plants were generated, how the mock plants were generated (using non-viruliferous aphids?) This is really important as these co-infected plants are one of the pivot material of the manuscript for both EPG and virus detection in the plant tissues. For example, it is not clear whether both viruses were inoculated at once or in subsequent inoculations, or if same aphid acquired sequentially both viruses and then inoculated them in the test plants. Authors should provide a detailed description of how these plants were generated, number of aphids used, age of the healthy plants, age of the plants when used in the experiments, etc

We modified it (Lines 157ff) and it reads now *“Non-viruliferous aphids were allowed a 24 h acquisition access period on BYV-infected plants or on purified BChV particles. Then they were transferred to 2-week-old healthy sugar beet plants for a 3-day inoculation access period. About 10 aphids of each condition were placed simultaneously on a plant for mono- and co-infection. Aphids were then removed manually. Inoculated and healthy, non-inoculated control plants were maintained in a growth chamber under controlled conditions (22-25 °C and 16/8 h light/dark photoperiod) for three weeks before being analyzed by DAS-ELISA to verify infection.”*

Note that we did not use mock-inoculated plants as controls, but healthy not-infested plants. This might have introduced some bias, but adding a fifth condition would have been very time-consuming. Further, our main focus was to compare mono- and co-infection and not mock *vs* these.

Results/discussion

Lines 310: **‘However, aphids spent significantly more time (~ one extra hour) ingesting phloem sap (E2 phase) on BYV mono-infected than on healthy or BChV-infected plants.’**

For me, that is a quite interesting result. One could suggest that a persistently infected plant could maybe promote sap phloem ingestion. However, this is only observed for the semipersistent virus (at least 1.5 h, which is a lot in a 8h recording). Even though that is not the focus of the work, I think it would be a quite small discussion about this fact.

We discuss this and added in Line 420 (discussion): *“Expectations are that the effect of virus localization in the host on vector behavior is more important than its persistence in the vector (K. E. Mauck et al., 2018). We confirm this here by showing that semi-persistent BYV and persistent BChV, both phloem-limited and acquired by aphids during phloem sap ingestion, induce similar host-mediated effects on vector behavior, i.e. we observed a neutral effect of BChV and a positive effect of BYV on phloem sap ingestion (Fig. 2).”*

In fact, the feeding times observed for E5 correlate quite well with the transmission rates obtained. Even though more time was spent in E2 in the co-infected plants than in the single-BChV-infected plants, this even reduced the acquisition from the double-infected plants. This was not observed for BYV (the other way around, but this did not affected transmission for the semipersistent). I think authors should go deeper on this interesting fact in the discussion and supporting on previous works.

We discussed this in our conclusion and mentioned that for BYV transmission the shorter E2 phase in co-infected plants (compared to plants mono-infected with BYV) is still sufficiently long for optimal BYV acquisition and transmission, and that increased phloem ingestion on co-infected plants (compared to plants mono-infected with BChV) indicates that it is not the aphid behavior that explains the drop in transmission of BChV from co-infected plants. For more clarity, we changed the text. Please see our response below to your comment (Lines 440: **We assume that...**)

The stage in which the virus source plants were used for the transmission experiments was the same as that used for the EPG experiments?

We used plants of the same age of infection for all experiments. Please see the Methods section.

Also, some variables are missing significance. Please, add it to all bars

Letters indicating the statistical significance have been added to all the error bars.

Lines 440: **We assume that under our experimental conditions, the 3 h sap ingestion observed for aphids feeding on co-infected plants is sufficient to charge aphids maximally with BYV and that for this reason.** That is true for BYV, but why the same effect is not observed for mixed- infected on the transmission of BChV

BYV is a non-circulative semi-persistent virus and BChV is a circulative persistent one. We assume that saturation kinetics are different for the two viruses since the binding sites may be limited for BYV, which is supposed to anchor to external stylet receptors, whereas BChV accumulates in the aphid hemolymph after intestinal uptake and the intestinal receptors may not be prone to saturation. Because prolonged phloem feeding on co-infected plants did not increase transmission of BChV (we observed the opposite effect), feeding activity and transmission efficiency are in this case not coupled. For better understanding, we modified the text correspondingly that now reads starting Line 433: “*Since BChV should be acquired from the phloem sap, longer phloem sap ingestion should increase acquisition (Gray et al., 1991; Prado & Tjallingii, 1994), because BChV probably uses, like related poleroviruses (Mulot et al., 2018), receptor-coupled endocytosis for intestinal uptake, a non-saturable mechanism because of replacement or recycling of used receptors. However, our results showed that despite aphids tending to ingest phloem sap for longer durations on co-infected plants than on mono-infected ones, transmission of BChV dropped. This indicates that the decreased transmission of BChV cannot be explained by aphid feeding behavior.*

Interestingly, for BYV, transmission from co-infected plants was not affected although aphids tended to feed less on co-infected plants than on mono-infected plants. We assume that under our experimental conditions, the 3 h of phloem sap ingestion observed for aphids feeding on co-infected plants is already sufficient to charge aphids maximally with BYV, since there is probably limited binding capacity in the aphid foregut and more specifically in the cibarium, where BYV could, like a related closterovirus, citrus tristeza virus, accumulate (Killiny et al., 2016; Ponsen, 1972). This is in line with the results by Jiménez and co-workers who demonstrated that maximum acquisition of BYV was obtained after 3 to 6 h of phloem sap ingestion, with no significant increases in transmission rates observed for longer sap ingestion periods (Jiménez et al., 2018). Taken together, we have no evidence that the altered aphid feeding behavior observed on co-infected sugar beets is the cause of the lower BChV transmission from these plants.”

Lines 484: At this stage, the exact mechanism of interaction between BChV and BYV is unknown. Because BChV should be acquired predominantly from the phloem sap and only to a small extent from phloem cells (Prado & Tjallingii, 1994), we propose that co-infection limits BChV release from companion cells into the sieve tubes (Fig. 7).

Authors could not support their results based on EPG nor virus titer or distribution and suggested that the decrease in the BChV transmission in the co-infected cells could be explained due to interaction between viruses with the cell. In my opinion, this assumption is not well supported by the results obtained in the manuscript. It could a nice way to find an explanation to the decrease in transmission observed, however, there are no clear results supporting that hypothesis. Probably authors should be a bit cautious about this assumption, just referring it as a likely fact explain the results, among other possible reasons behind, or dealing with it in a more extensive paragraph if associating that hypothesis with published bibliography on the topic.

We weakened our hypothesis because it is true that we have barely any evidence for it. It reads now starting Line 482“*Since no other notable differences were detected in co-infected vs mono-infected plants, we propose that virus-virus interactions in co-infected cells might have caused the drop in BChV transmission. However, the mechanisms of interaction between BChV and BYV remain unknown at this stage. Because BChV should be acquired predominantly from the phloem sap and only to a small extent from phloem cells (Prado & Tjallingii, 1994), one hypothesis is that co-infection limits BChV release from companion cells into the sieve tubes (Fig. 7). This would result in a lower BChV accessibility to aphids, compared to BChV mono-infected plants.*”

Response to reviewer 3:

Thanks for evaluating our work.

- L. 41. Authors define "co-infection, multi-infection or mixed infection" as simultaneous presence of more than one pathogen in a given host. However, later in the same paragraph (l. 47), they differentiate correctly co-infection and super-infection as the results of the simultaneous or sequential arrival of the two pathogens, respectively. I agree with this idea to separate both concepts, therefore it might be better not to use co-infection in l. 41.

We deleted “co-infection” in line 41.

Later it will be a good idea to state at the beginning of the results section that all the experiments were performed with co-infected plants (as described in methods, l. 157-163).

Also on demand of reviewer 2, we changed the title (Line 157) to “*Generation of mono- and co-infected source plants for subsequent experiments*” replaced “*mixed infection*” to clarify that creation of source plants is different from aphid transmission experiments. We think therefore also that it is not necessary to repeat this at the beginning of the results section.

- L. 321. Correct misspelling of the statistical method employed: Tukey HSD

Thanks for noting the spelling error. We corrected it.

- L. 502. Add a short statement here to clarify that the case of study was not unknown to science, although it can be said that it was not previously addressed in the same depth including the transmission perspective.

We changed the sentence and added two references to reflect the fact that closterovirus/polerovirus co-infection has been studied before, but not on the cellular or behavioral level: “*a combination that has been studied before (Hossain et al., 2021; Wintermantel, 2005), but not in such depth.*”

- In Figure S1 is not necessary to plot the "expected" transmission rate, which is calculated theoretically from the data already presented in figure 1. Fig S1 can be better replaced by a table, or to provide the experimental data in the main text (l. 292).

We added the statistical data (Line 292) to the main text, but decided to keep Figure S1. The passage reads now: “*Aphids feeding on co-infected plants transmitted both viruses together at 19% transmission rate, which was significantly lower than expected, based on the virus transmission efficiency of each virus (36% expected co-infection rate, calculated as the product of the transmission rates using mono-infected source plants; p-value= 0.029; n=75, five independent experiments; df=1; chi-squared (χ^2)).*”

The data provided is solid, and only a certain speculation is leaved for the mechanism of virus-virus interaction. Considering that this interaction might be indirect/more complex than a testable binding of the two virions, and it might involve other viral products, the discussion of the case (l. 475-491) is quite fair.

We cooked down our conclusions. It reads now starting Line 482 “*Since no other notable differences were detected in co-infected vs mono-infected plants, we propose that virus-virus interactions in co-infected cells might have caused the drop in BChV transmission. However, the mechanisms of interaction between BChV and BYV remain unknown at this stage. Because BChV should be acquired predominantly from the phloem sap and only to a small extent from phloem cells (Prado & Tjallingii, 1994), one hypothesis is that co-infection limits BChV release from companion cells into the sieve tubes (Fig. 7). This would result in a lower BChV accessibility to aphids, compared to BChV mono-infected plants.*” See also our response to reviewer 2.

Re: Spectrum01115-24R1 (Interplay between a poliovirus and a closterovirus decreases aphid transmission of the poliovirus)

Dear Dr. Martin Drucker:

Your manuscript has been accepted, and I am forwarding it to the ASM production staff for publication. Your paper will first be checked to make sure all elements meet the technical requirements. ASM staff will contact you if anything needs to be revised before copyediting and production can begin. Otherwise, you will be notified when your proofs are ready to be viewed.

Sincerely,
Clinton Jones
Editor
Microbiology Spectrum

Reviewer #3 (Comments for the Author):

The answers of the authors have solved the few minor issues found in the previous version, and I am fully satisfied with the contents and discussion of the new revised manuscript.